# The Multi-Stage Drawing Process of Zinc-Coated Medium-Carbon Steel Wires in Conventional and Hydrodynamic Dies

**DOI:** 10.3390/ma13214871

**Published:** 2020-10-30

**Authors:** Maciej Suliga, Radosław Wartacz, Marek Hawryluk

**Affiliations:** 1Faculty of Production Engineering and Materials Technology, Czestochowa University of Technology, 19 Armii Krajowej Av. 42-200 Czestochowa, Poland; imitm@wip.pcz.pl; 2Department of Metal Forming and Metrology, Wroclaw University of Science and Technology, 5 Lukasiewicza str., 50-371 Wrocław, Poland; marek.hawryluk@pwr.edu.pl

**Keywords:** wire, zinc coating, hydrodynamic die, drawing speed, surface, properties

## Abstract

This paper discusses experimental studies aiming to determine the effect of the drawing method on the lubrication conditions, zinc coating mass and mechanical properties of medium-carbon steel wires. The test material was 5.5 mm-diameter galvanized wire rod which was drawn into 2.2 mm-diameter wire in seven draws at a drawing speed of 5, 10, 15, 20 and 20 m/s, respectively. Conventional and hydrodynamic dies with a working portion angle of α = 5° were used for the drawing process. It has been shown that using hydrodynamic dies in the process of multi-stage drawing of zinc-coated wire improves the lubrication conditions, which leads to a reduction in friction at the wire/die interface. As a consequence, wires drawn hydrodynamically, as compared to wires drawn conventionally, are distinguished by a thicker zinc coating and better mechanical and technological properties.

## 1. Introduction

External friction between the wire and the drawing die is among the factors which determines the conditions of the top wire layer [1,2,3,4,5]. Among the known methods of improving lubrication conditions and reducing friction is the application of hydrodynamic dies, the so-called pressure dies in the drawing process. The drawing process performed with the use of hydrodynamic dies consists in introducing a lubricant into the resistance die through a narrow slot. As a result, the lubricant becomes a liquid, successively increasing the pressure in the die, where the greatest value is achieved at the contact of the working die with the wire. The generated high pressure not only separates the friction surfaces of the wire and the working die but also contributes to the deformation by changing the diameter and shape of the drawn wire [6,7,8].

In the case of drawing zinc-coated wire, this layer consists of zinc coatings and a steel substrate [9]. In hot-dip zinc coating, the properties of the zinc coating depend largely on the galvanizing technology, including the composition and temperature of the bath and the immersion time [9,10,11,12]. In contrast to drawing uncoated wire, a very high variation in physicochemical properties between individual wire layers exists in the surface layer of zinc-coated wire at a depth of approximately 100 µm. In the multi-stage drawing process, the top surface layer is intensively heated up. This, in turn, is expected to influence the conditions of deformation of the soft zinc coating on the hard steel core, including wire surface roughness. The formation of the wire topography also depends on the drawing technology. The literature on the zinc-coated wire drawing technology lacks information concerning the surface roughness of zinc-coated wire after the multi-stage drawing process at high drawing speeds (above 5 m/s). Nevertheless, the literature data on single-stage drawing at a speed of up to 2 m/s show that the angle and the drawing method influence both the top wire layer and the zinc coating surface roughness [13,14,15,16].

## 2. Materials and Methods

The test material was 5.5 mm-diameter galvanized steel wire rod (Drumet, Wloclawek, Poland). The examination of the microstructure of the coatings was made with the use of an S-3400 N-type Hitachi scanning microscope equipped (Hitachi, Tokyo, Japan) with an Energy Dispersion Spectroscopy X-ray spectrometer. An accelerating voltage of 25 kV was used. Figure 1 shows the structure of the zinc coating before and after the drawing process (wire rod with a diameter of 5.5 mm).

The SEM investigation showed that after the hot-dip galvanizing process of the wire rod, the zinc coating consists of an outer layer and a diffusion layer containing intermetallic phase layers, Figure 1a. The process of drawing wires with a diameter of 2.2 mm caused, regardless of the drawing technology, a more than two-fold decrease in the thickness of the zinc coating, Figure 1b.

The wire rod was drawn into 2.2 mm-diameter wire on a multi-stage drawing machine in seven draws at a drawing speed of 5, 10, 15, 20 and 20 m/s, respectively (Table 1 and Table 2). Conventional dies and hydrodynamic dies with a working portion angle of α = 5° were used for the drawing process.

The examination of the surface topography of the zinc-coated wires after the drawing process was carried out on a Form Talysurf 50e profilometer (Taylor Hobson, Leicester, England). Figure 2 shows an example of a profilogram of the surface texture of 2.20 mm-diameter wire.

Figure 2 shows an example of a profilogram of the surface texture of 2.20 mm-diameter wire. To illustrate the variations in the wire surface texture, the profile height and profile deviation parameters were used for the analysis. The average line (Figure 2) is understood as a profilogram line which divides the surface into two parts in such a manner that the surface calculated using the integral is equal to zero. The circle in the center of the profilogram of the geometric structure of the wire surface represents the center of the wire roughness measurement on the analyzed length, while the numbers on the x-axis represent the current position of the measuring head of the device.

The profile height parameter describes the height of the profile irregularities using linear dimensions perpendicular to the average line or the arithmetic mean. The explanation of the profile height can be represented with the use of the following parameters:

R_p_—the maximum height of the profile elevation above the value of the average line within the measuring length under examination;

R_v_—the maximum depth of the profile below the average line within the measuring length;

R_t_—the total value of the profile depth and profile height within the measuring length under examination:(1)$$R_{t}=R_{p}+\left|R_{v}\right|$$

R_z_—the arithmetic mean of the absolute values of the heights of the five largest elevations and the five largest depressions of the roughness profile:(2)$$R_{z}=\frac{\sum_{i=1}^{5}\left|y_{\mathrm{pi}}\right|+\sum_{i=1}^{5}\left|y_{\mathrm{ri}}\right|}{5}$$

The profile deviation parameters describe the deviation of the profile in the direction perpendicular to the average line, where the basis for the calculation of the measure of dispersion is the assumption of mathematical statistics. They are calculated as follows:

R_a_—the mean of the profile deviations from the average line:(3)$$R_{a}=\frac{1}{L}\overset{L}{\underset{0}{\int }}\left|y\left(x\right)\right|\mathrm{dx}$$

R_q_—the quadratic mean of the values of profile deviations from the average line within the examined measuring length:(4)$$R_{q}=\sqrt{\frac{1}{1}\overset{1}{\underset{0}{\int }}y^{2}\left(x\right)}\mathrm{dx}$$

The examination of the zinc-coated wire roughness measurements was carried out for finished 2.2 mm-diameter wires. For the structural examination of the top wire layer geometry, five elementary segments, each of a measuring length of 0.8 mm, were used. The wire roughness measurements were performed for the longitudinal direction to the drawing direction.

## 3. Results and Discussion

Lubrication conditions. To determine the effect of the drawing method on the lubrication conditions, ten 100 mm-long specimens were taken for each drawing variant. After the specimens were weighed on a laboratory balance, the lubricant layer was removed with the use of sodium hydroxide (NaOH) and technical acetone. After the specimens had completely dried up, they were weighed again. From the mass difference, the amount of lubricant on the wire surface was determined. The test results are presented in Figure 3 and Figure 4 for conventional dies (K) and hydrodynamic dies (H), respectively.

The test results illustrated in Figure 2 and Figure 3 confirm the significant effect of the drawing method on the lubrication conditions. The data in Figure 2 and Figure 3 show that using hydrodynamic dies in the zinc-coated wire drawing process improves the lubrication conditions, while the higher the drawing speed and the greater the total reduction, the larger the differences. For wires drawn hydrodynamically at drawing speeds of 5 and 20 m/s, as compared to wires drawn conventionally, a lubricant amount larger by 26.6% and 114.8%, respectively, was recorded on the wire surface. The improvement in the lubrication conditions when drawing in hydrodynamic dies can be explained by more favorable conditions for the deformation of the top layer of the zinc-coated wire.

Analysis of the surface topography of zinc-coated wires. In drawing processes, aside from the die geometry and the drawing speed, the method of drawing has a significant influence on the formation of the wire surface. Therefore, a comparison of the surface roughness of zinc-coated wires drawn in conventional dies and hydrodynamic dies, respectively, was made in the study.

Based on the results, graphs were plotted to illustrate the effect of the drawing method on the surface texture parameters of 2.2 mm-diameter zinc-coated wires—see Figure 5, Figure 6, Figure 7, Figure 8, Figure 9 and Figure 10.

The analysis of the surface roughness of wires drawn conventionally and hydrodynamically has shown that using hydrodynamic drawing dies reduces the surface roughness of zinc-coated wire. At a drawing speed of 20 m/s, the difference between the drawing variants is as follows: 12.5% R_v_, 20.4% R_v_, 15.8% R_t_, 11.3% R_z_, 18.7% R_a_ and 44.6% R_q_, respectively. Using hydrodynamic dies in the drawing process creates more favorable conditions for the deformation of the top layer of the zinc-coated wire. The better lubrication conditions, as confirmed by a greater amount of the lubricant after the drawing process (Figure 3), have contributed to a reduction in friction and the wire top surface temperature. In contrast to drawing uncoated wire, significant changes to the physicochemical properties of the zinc coating occur in the case of drawing zinc-coated wire. Using hydrodynamic dies diminishes the adverse effect of the drawing speed on the softening of the thin zinc coating in the die. Hence, wires drawn by the hydrodynamic method have a smoother surface.

Examination of zinc coating mass. The mass of the zinc coating was determined by the gravimetric method in accordance with the applicable standard PN-EN 10244-1. Then, the change in the zinc coating mass after drawing in conventional dies and hydrodynamic dies was compared. The test results are illustrated in Figure 11 and Figure 12.

The analysis of the surface roughness of wires drawn conventionally and hydrodynamically has shown that using hydrodynamic drawing dies reduces the surface. It can be seen from Figure 11 and Figure 12 that the drawing method significantly influences the variation of the zinc coating mass on the wire after the drawing process. Using hydrodynamic dies in the multi-stage zinc-coated wire drawing process favorably influences the drawing conditions and the zinc coating mass. While at a drawing speed of 5 m/s, the difference in the zinc mass between the wires drawn conventionally and hydrodynamically amounts to 3.8%, at a drawing speed of 20 m/s, it already exceeds 47%. It has also been found that the difference in the zinc mass between variants K and H increases with the increase in total reduction. At a reduction of Gc = 84% and a speed of 10 m/s, this difference is approximately 15%.

When analyzing the mass of the coating (based on standard PN-EN 10244-2), which should be present on the wire when it is categorized into the respective class, it can be found that, for a drawing speed of 5 m/s, the coatings on wires drawn either conventionally or hydrodynamically are situated in class AB, whereas the coating in the hydrodynamic method is thicker by 3.8% compared to that in the conventional method. At a drawing speed of 10 m/s, both variants still hold class AB, while the difference between the variants being almost 15% of the zinc mass on the other wire.

A significant difference of 30% in the zinc coating mass between the variants under analysis caused variant K (conventional dies) to be categorized into a lower class—i.e., B, at a drawing speed of 15 m/s, while variant H (hydrodynamic dies) achieved class AB. The increase in the drawing speed from 15 to 20 m/s did not change the class of the coatings.

In the hydrodynamic method, the mass of the remaining zinc coating is dependent on the lubricant pressure created in the die during the drawing process, which directly influences the friction conditions and the wire heating. Sufficiently high lubricant pressure in the hydrodynamic die allowed relatively good lubrication conditions to be maintained. This, in turn, contributed to a lowering of wire temperature. In consequence, the wires drawn hydrodynamically were distinguished by a thicker zinc coating. Unlike the conventional method, no such negative effect of high drawing speeds on the zinc coating thickness was noted for the hydrodynamic method. Therefore, the wires drawn in hydrodynamic dies at a drawing speed of 20 m/s had the coating class AB.

Testing for mechanical and engineering properties. Tests aiming to determine the mechanical and engineering properties of the wire were performed in accordance with standard PN-EN 10218-1:2012 on a Zwick/Z100 testing machine and on a wire twisting and bending test device. Wires of a diameter of 2.2 mm were subjected to testing to determine the yield strength, R_0.2_; ultimate tensile strength, R_m_; uniform elongation, A_r_; total elongation, A_c_; reduction in area, Z; the number of twists, N_t_; and the number of bends, N_b_. The results of the mechanical and technological tests are represented in Figure 13, Figure 14, Figure 15, Figure 16 and Figure 17. Figure 13 shows that using hydrodynamic dies results in a reduction in the mechanical properties of zinc-coated steel wire. Wires drawn in hydrodynamic dies, as compared to wires drawn conventionally, showed a yield point decrease, on average, of 2.7% and an ultimate tensile strength decrease of 2.3%.

The lower strain hardening of the hydrodynamically drawn wires can be linked with more favorable conditions of lubrication as well as wire and zinc coating deformation in the drawing process. Hence, the wires drawn by the hydrodynamic method exhibited better plastic properties, as confirmed by the data illustrated in Figure 14 and Figure 15.

The data in Figure 14 and Figure 15 show that, with the increase in the drawing speed, the differences in the plastic properties between the conventionally and hydrodynamically drawn wires increase and exceed 10% at a drawing speed of 20 m/s. Wires drawn hydrodynamically at a drawing speed of 20 m/s, as compared to wires drawn conventionally, show reductions in uniform elongation and total elongation values of 10.6% and 9.9%, respectively, as well as a reduction in the area of 11.8%. Using hydrodynamic dies in the zinc-coated wire drawing process also has a favorable effect on the technological properties, such as the number of twists, N_t_, and the number of bends, N_b_. This is confirmed by the data shown in Figure 16 and Figure 17.

Using hydrodynamic dies largely removes the adverse effect of the drawing speed on the technological properties of the wire. In the drawing speed range of 5–20 m/s, using hydrodynamic dies caused an increase in the number of twists, N_t_, by 1% to 7.2% and an increase in the number of bends, N_b_, by 1.4% to 8.7%, compared to conventional dies. Therefore, the use of hydrodynamic dies in the zinc-coated wire drawing process enables not only a thicker zinc coating on the finished wire but also better mechanical and technological properties to be obtained, especially at drawing speeds not exceeding 15 m/s.

Analysis of residual stresses in steel wires. Residual stress is one of the basic parameters significantly affecting the quality of the steel wire. Mechanical methods are the best methods for determining the residual stresses in wires, allowing quick stress measurements. The analysis of the influence of the drawing method on the residual stresses in end wires with a nominal diameter of 2.2 mm was determined based on the method of longitudinal wire cutting, known in the literature as the Schepers–Peiter method. This method involves cutting the wire to a certain length and measuring the deflection value at the end of the wire. The end wires with a nominal diameter of 2.2 mm were cut on a wire EDM machine to the length l = 44 mm (ratio l/d = 20). Schepers–Peiter [16] proposed that from the equation of the moments of forces acting on the cut wire, the following expression is obtained for the circular-symmetric distribution to determine the stresses on the outer surface of the wire:(5)$$\sigma_{w}=1.3176\cdot E\cdot R\cdot \frac{h}{l^{2}}$$
where:σ_w_—longitudinal residual stress on the wire surface, MPa;E—Yung’s module, MPa;R—wire radius, mm; h—parting of the wire ends, mm;l—cutting length, mm.

The results of type I residual stresses (average values of 10 wires) are presented in Table 3. In order to better analyze the obtained test results, the percentage differences between the analyzed drawing variants are also presented.

On the basis of the results presented in Table 3, diagrams were prepared showing the influence of the drawing method on the first type residual stresses, Figure 18.

Based on the data presented in Figure 18, it can be concluded that the drawing method significantly affects the residual stresses in steel wires. The use of hydrodynamic dies in the drawing of galvanized steel wires causes a decrease in residual stress, especially in the speed range of 5–15 m/s. With the increase in the drawing speed, the differences in the values of longitudinal residual stresses between the wires drawn with the conventional and hydrodynamic methods decreased, and were: 42.5%, 39.6%, 27.7% and 3.3%, respectively

Lower values of residual stresses in hydrodynamically drawn wires should be seen with the occurrence of smaller deformations for this variant. Separating the friction surfaces of the wire and die with a sufficiently thick layer of lubricant allowed a significant decrease in the friction coefficient, and this, in turn, reduced the deformation resistance. As a consequence, the wires drawn with the hydrodynamic method are characterized by a lower heterogeneity of the deformation on the wire cross-section, hence the decrease in residual stresses for this drawing variant. Lower residual stresses in hydrodynamically drawn wires can also be seen in a more uniform temperature distribution, because the lower the friction coefficient, the lower the temperature of the wire surface layer heated by friction.

## 4. Conclusions

The carrying out of the process of drawing zinc-coated wire in hydrodynamic dies enables an improvement in the lubrication conditions and reduces the friction at the wire/die interface. Therefore, a larger amount of zinc has been applied on the wire surface for wire drawn hydrodynamically, as opposed to wire drawn conventionally.

Using hydrodynamic dies partially removes the adverse effect of high drawing speeds (above 10 m/s) on the softening of the thin zinc coating in the die. In consequence, wire drawn by the hydrodynamic method has a smoother surface.

Wires drawn in hydrodynamic dies in the drawing speed range of 5-20 m/s showed, on average, a yield point decrease of 2.7% and an ultimate tensile strength decrease of 2.3%, as compared to wires drawn conventionally. The lesser strain hardening of hydrodynamically drawn wires can be associated with more favorable conditions of lubrication as well as wire and zinc coating deformation in the drawing process. Therefore, wires drawn by the hydrodynamic method were distinguished by better plastic properties.

Using hydrodynamic dies largely removes the adverse effect of the high drawing speeds on the engineering properties of the wire. Therefore, hydrodynamically drawn wires showed a number of twists greater by 1% to 7.2% and a number of bends greater by 1.4% to 8.7%.

The application of hydrodynamic dies in the drawing process of galvanized steel wires causes a decrease in residual stress, especially in the speed range 5–15 m/s. The separation of the friction surfaces of the wire and the hydrodynamic drawing die with a sufficiently thick layer of lubricant allowed a significant decrease in the friction coefficient, deformation resistance and deformation heterogeneity on the wire cross-section. In consequence, the wires drawn hydrodynamically are characterized by lower residual stresses.

The obtained investigation results can be used in the design of technologies for multi-stage drawing of zinc-coated steel wire. According to the authors, in the process of drawing galvanized wires, hydrodynamic dies should be used in all drafts.

## Figures and Tables

**Figure 1 materials-13-04871-f001:**
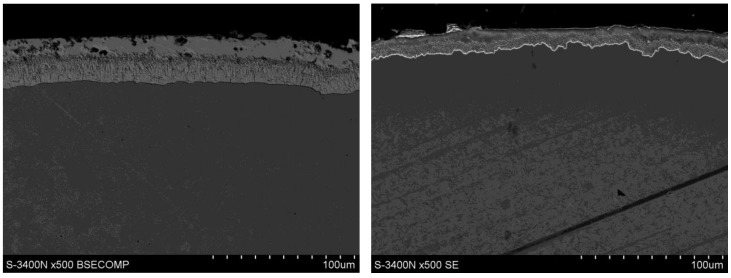
Coat cross-section (SEM) microstructure for: (**a**) wire rod, (**b**) 2.2 mm wire drawn in conventional dies.

**Figure 2 materials-13-04871-f002:**
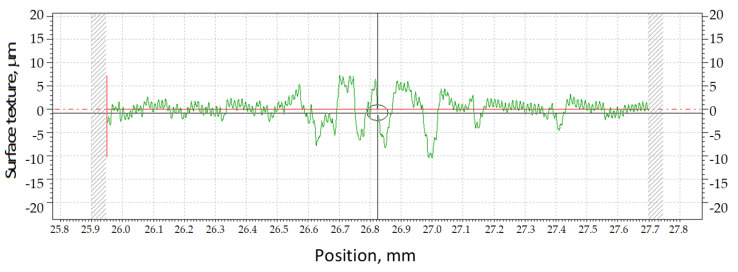
The surface texture of ϕ2.2 mm-diameter wire drawn in hydrodynamic dies.

**Figure 3 materials-13-04871-f003:**
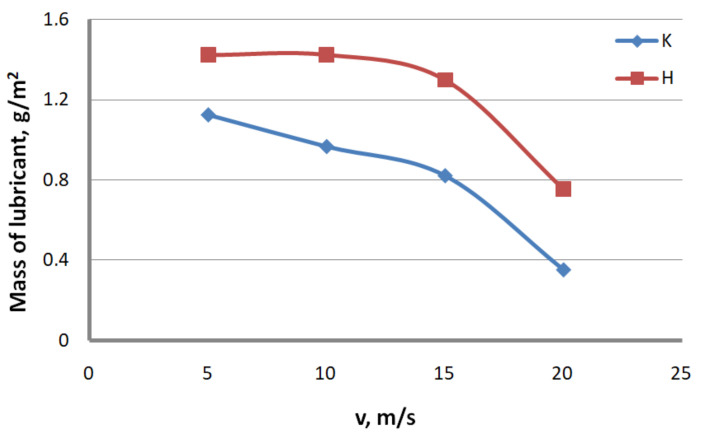
The effect of the drawing speed, v, on the lubricant mass on the surface of 2.2 mm-diameter wires drawn in conventional dies (K) and hydrodynamic dies (H), respectively.

**Figure 4 materials-13-04871-f004:**
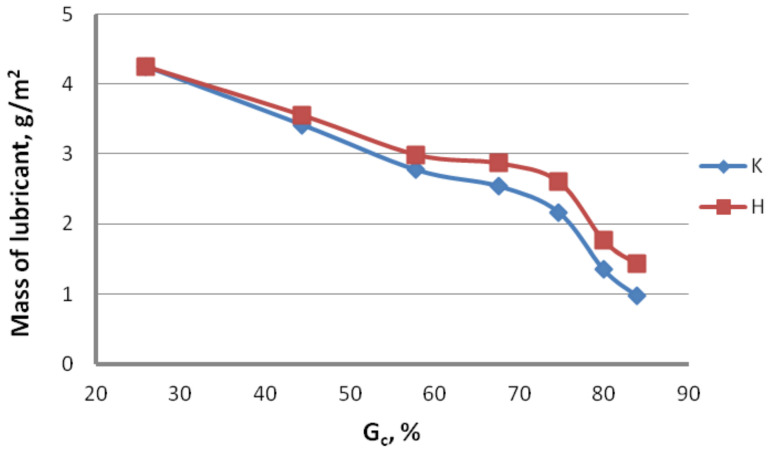
Variation in the lubricant mass on the surface of wires drawn in conventional dies (K) and hydrodynamic dies (H) as a function of total reduction (v = 10 m/s).

**Figure 5 materials-13-04871-f005:**
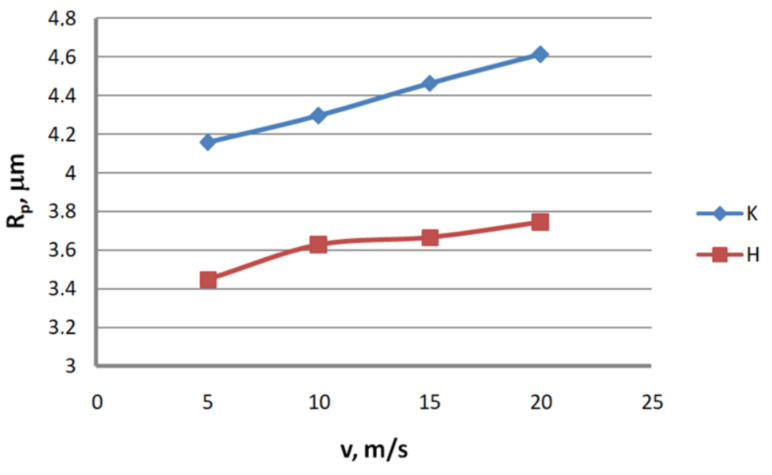
The effect of the drawing speed, v, on the surface profile height parameter, R_p_, for 2.2 mm-diameter wires drawn in conventional dies (K) and hydrodynamic dies (H), respectively.

**Figure 6 materials-13-04871-f006:**
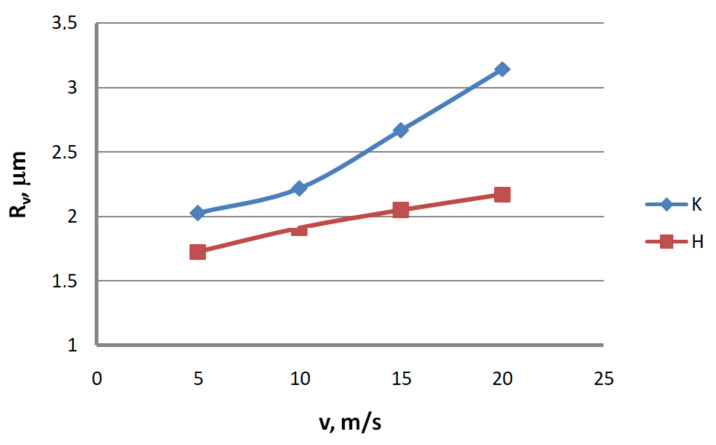
The effect of the drawing speed, v, on the surface profile height parameter, R_v_, for 2.2 mm-diameter wires drawn in conventional dies (K) and hydrodynamic dies (H), respectively.

**Figure 7 materials-13-04871-f007:**
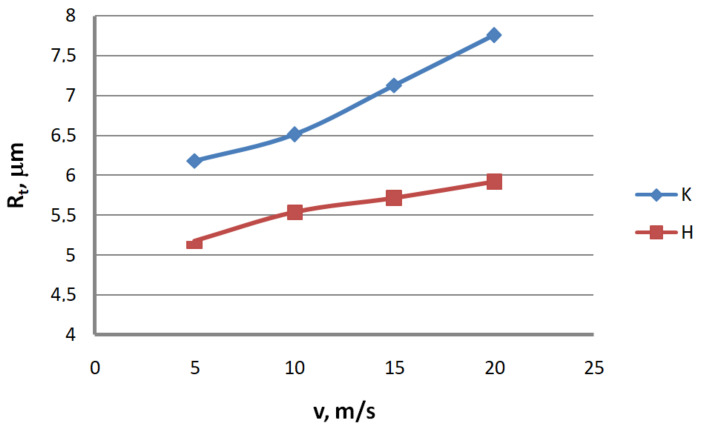
The effect of the drawing speed, v, on the surface profile height parameter, R_t_, for 2.2 mm-diameter wires drawn in conventional dies (K) and hydrodynamic dies (H), respectively.

**Figure 8 materials-13-04871-f008:**
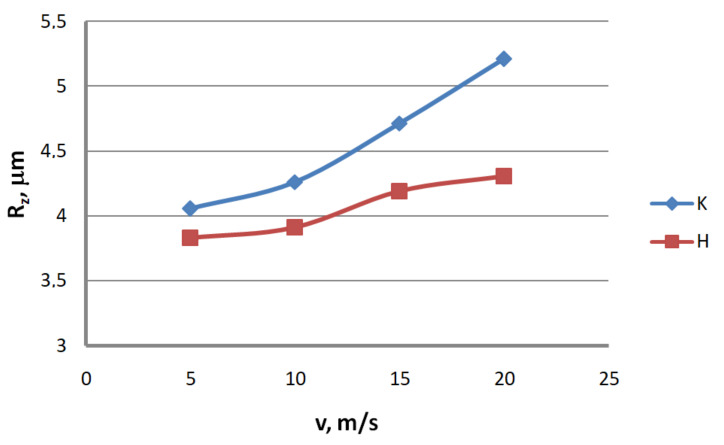
The effect of the drawing speed, v, on the surface profile height parameter, R_z_, for 2.2 mm-diameter wires drawn in conventional dies (K) and hydrodynamic dies (H), respectively.

**Figure 9 materials-13-04871-f009:**
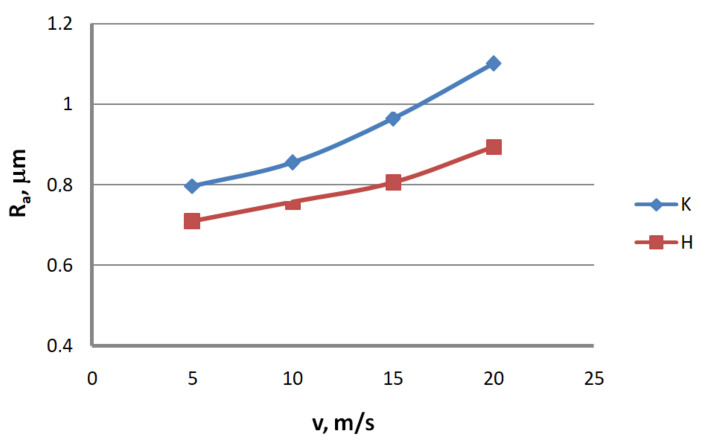
The effect of the drawing speed, v, on the surface profile deviation parameter, R_a_, for 2.2 mm-diameter wires drawn in conventional dies (K) and hydrodynamic dies (H), respectively.

**Figure 10 materials-13-04871-f010:**
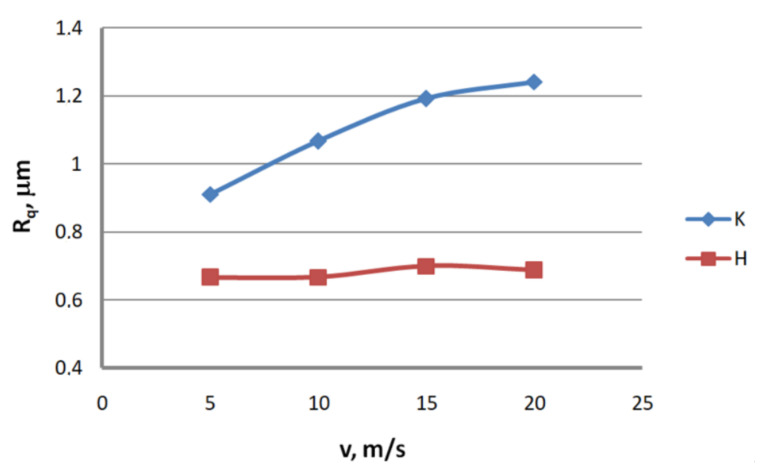
The effect of the drawing speed, v, on the surface profile deviation parameter, Rq, for 2.2 mm-diameter wires drawn in conventional dies (K) and hydrodynamic dies (H), respectively.

**Figure 11 materials-13-04871-f011:**
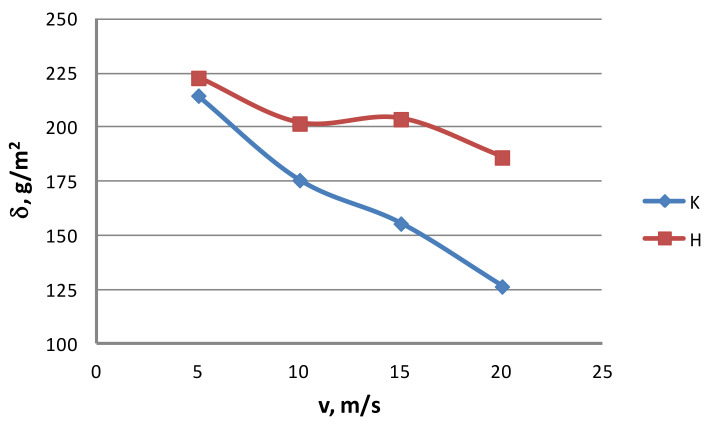
The effect of the drawing speed, v, on the zinc mass δ on the surface of 2.2 mm-diameter wires drawn in conventional dies (K) and hydrodynamic dies (H), respectively.

**Figure 12 materials-13-04871-f012:**
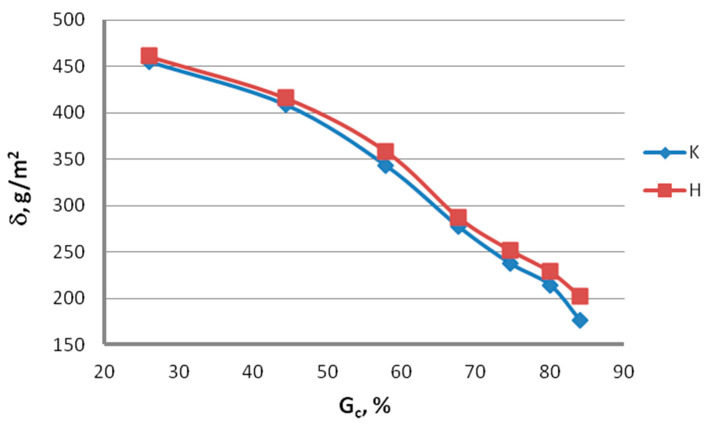
Variation in the zinc mass δ on the surface of wires drawn in conventional dies (K) and hydrodynamic dies (H) as a function of total reduction (v = 10 m/s).

**Figure 13 materials-13-04871-f013:**
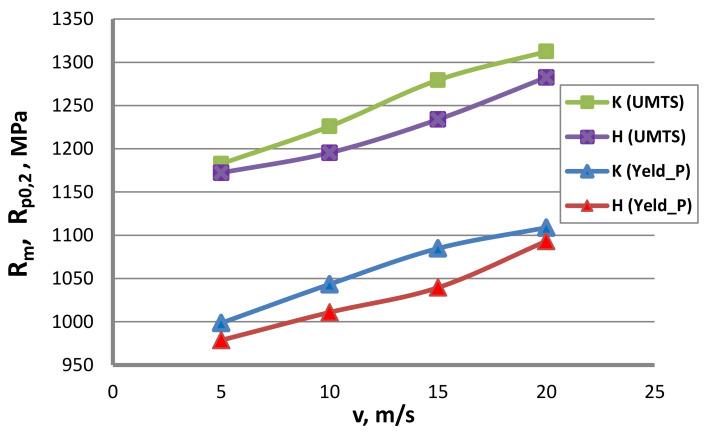
The effect of the drawing speed, v, on the yield strength (UMTS) and yield point, R_0.2_, of 2.2 mm-diameter wires drawn in conventional dies (K) and hydrodynamic dies (H), respectively.

**Figure 14 materials-13-04871-f014:**
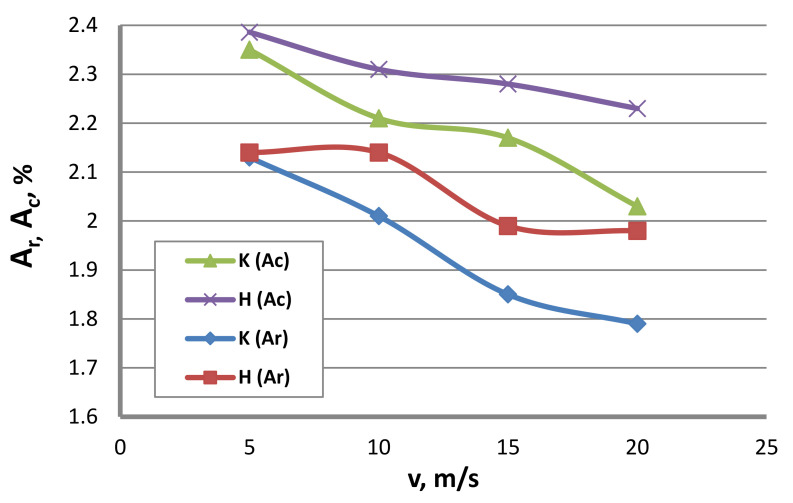
The effect of the drawing speed, v, on the uniform elongation, A_r_ and also on the total elongation Ac of 2.2 mm-diameter wires drawn in conventional dies (K) and hydrodynamic dies (H), respectively.

**Figure 15 materials-13-04871-f015:**
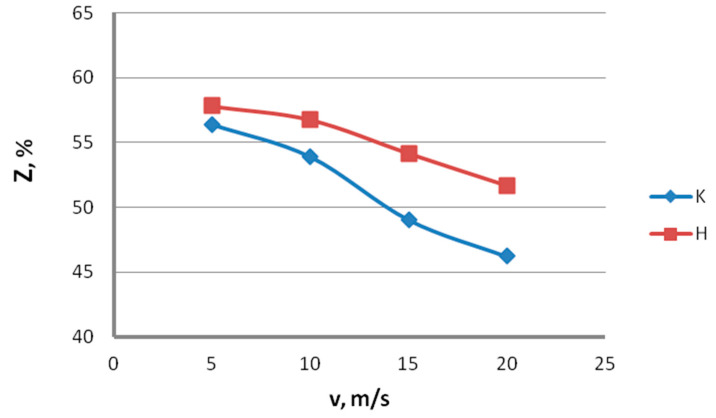
The effect of the drawing speed, v, on the reduction in area, Z, of 2.2 mm-diameter wires drawn in conventional dies (K) and hydrodynamic dies (H), respectively.

**Figure 16 materials-13-04871-f016:**
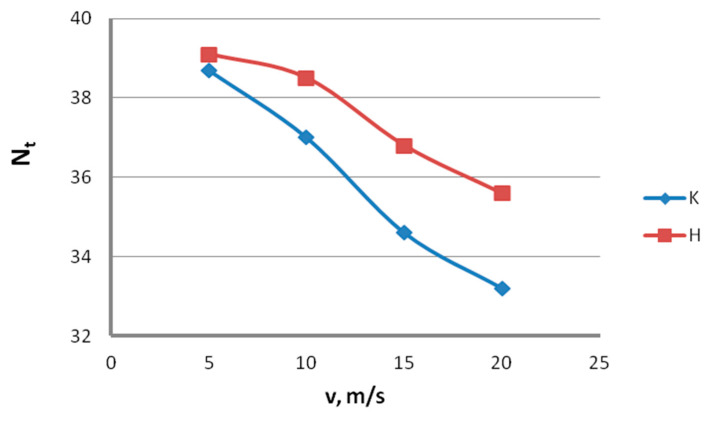
The effect of the drawing speed, v, on the number of twists, N_t_, of 2.2 mm-diameter wires drawn in conventional dies (K) and hydrodynamic dies (H), respectively.

**Figure 17 materials-13-04871-f017:**
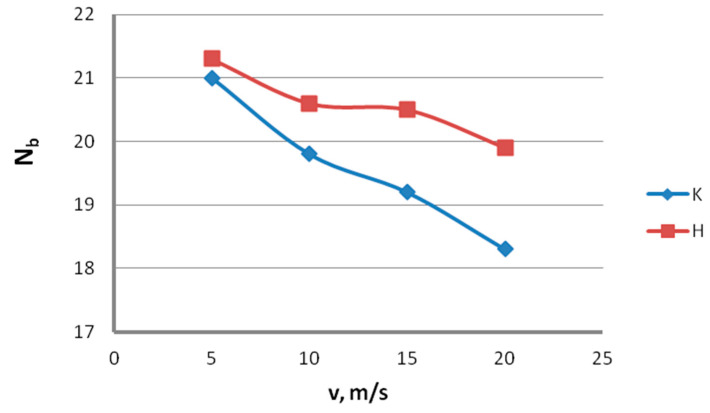
The effect of the drawing speed, v, on the number of bends, N_b_, of 2.2 mm-diameter wires drawn in conventional dies (K) and hydrodynamic dies (H), respectively.

**Figure 18 materials-13-04871-f018:**
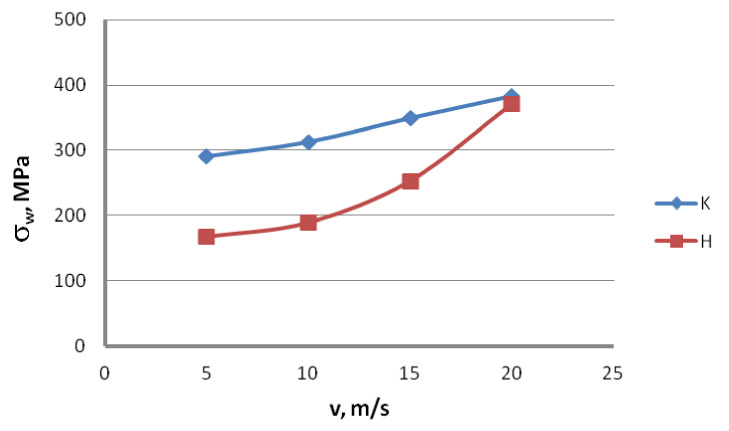
The effect of the drawing speed, v, on the longitudinal residual stresses σ_w_ of 2.2 mm-diameter wires drawn in conventional dies (K) and hydrodynamic dies (H), respectively.

**Table 1 materials-13-04871-t001:** Distribution of single reductions, G_p_, and the total reduction, G_c_.

Draw No.	0	1	2	3	4	5	6	7
ϕ, mm	5.50	4.73	4.10	3.57	3.13	2.77	2.46	2.20
G_p_, %	-	26.04	24.86	24.18	23.13	21.68	21.13	20.02
G_c_, %	-	26.04	44.43	57.87	67.61	74.64	79.99	84.00

**Table 2 materials-13-04871-t002:** Drawing speed, v [m/s], at individual drawing drafts.

Draw No.						
1	2	3	4	5	6	7
Drawing speed, v [m/s], at individual drafts						
1.06	1.43	1.90	2.47	3.15	4.00	5
2.12	2.86	3.80	4.94	6.31	8.00	10
3.17	4.28	5.70	7.41	9.46	12.00	15
4.22	5.70	7.59	9.88	12.62	16.00	20

**Table 3 materials-13-04871-t003:** Results of the residual stress tests performed by the longitudinal cutting method for the final wires with a nominal diameter of 2.2 mm drawn in conventional dies (K) and in hydrodynamic dies (H).

v, m/s	Die Angle α, °		(K-H)/K, %
	K	H	
	Residual Stress σ_w_, MPa		
5	290.6	167.1	42.5
10	312.7	189.0	39.6
15	349.4	252.7	27.7
20	383.3	370.6	3.3

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
