# Peer review of "The Multi-Stage Drawing Process of Zinc-Coated Medium-Carbon Steel Wires in Conventional and Hydrodynamic Dies"

_materials, 2020, doi:10.3390/ma13214871_

Round 1

Reviewer 1 Report

  1. the paper shows interesting results regarding the cold drawing of galvanized middle-carbon steel wire. The experiments are well done and cleared reported.
  2. this is it, through, as there is little interpretation of the results, either in terms of the mechanics of continuous media or of metallurgical interpretation, based on metallographic observations, of what happens to the zinc layer and sublayers, during wire drawing. For example, the following thesis contains much helpful information: Guillaume VEGA, Optimisation de la mise en forme par tréfilage : approche expérimentale, modélisation et simulation numérique. 7/12/209, Lille, France. 
  3. the way the hydrodynamic die works is just touched upon - a few words and sentences, here and there (e.g."can be explained by more favourable conditions for the deformation of the top layer of zinc-coated wire"). By the way, a definition of an hydrodynamic die, at the beginning, would help the reader!
  4. in the conclusion, what is proposed for industrial implementation, based on the findings of this work, would be welcome! Convert 100% to dynamic dies? Only at the very end? 
  5. if you decide to flesh out the paper with more "theoretical" analysis, then, maybe, you can compact your results in less figures, each containing several graphs?
  6. check the quality of your English. I highlighted some words and expressions in the text, which would need a slight rewriting, copy attached. For example, at the beginning, please choose between zinc coat or zinc coating! 

Author Response

Thank You all for complex and accurate revisions of our manuscript. We admit, that there were several mistakes pointed in your reviews, which we found and corrected in current version of manuscript.

We admit that there were several mistakes pointed out in your reviews. All significant changes resulting from the comments and suggestions of both reviewers were marked in red in the revised text of the manuscript. In this letter, we responded to all the comments and we added statements of the changes we have made. We will be highly honored if you are willing to examine this manuscript again.

Reviewer 2 Report

The paper examines the question of the optimum drawing process for zinc-coated medium-carbon steel wires. The data are convincing. It’s shown that the use of a hydrodynamic die allows for a considerable improvement of the quality of the final product. The manuscript can be accepted for publication after several minor revisions listed below.

  1. Figure 4.

To begin with, it would be logical to move the sentence concerning the profilometer (lines 89-90) to Section 2, Materials and Methods.

Moreover, it should not be forgotten that the paper must be understandable for a wide audience and not only for scientists from exactly the same domain of research. The profile measurements need to be explained in more detail. In particular, in which direction is the “Position” measured? Is it the length direction? If yes, why is not the mean roughness uniform along the length, but displays a maximum near the vertical solid line? What is the meaning of this line? What is the meaning of the red line, of the circle in the center… Please, explain the measurements and develop the figure caption.   

  1. The text should be reread to remove misprints, clumsy phrasing, but also missing articles. Although the English is globally good, the articles need to be checked by a professional interpreter.

Some examples of misprints and clumsy phrasing are given below:

Lines 142,143. “With the increase in drawing speed, the difference between drawing variants under analysis increases, at the drawing speed 20 m/s it being:”

The sentence should be reformulated.

Lines 163-165. “The analysis of the surface roughness of wires drawn conventionally and hydrodynamically has shown that using hydrodynamic drawing dies reduces the surfaceIt can be seen from Figures 11-12…”

A word or a sentence is lost within the construction “surfaceIt”.

Lines 228, 229. t and b should be indices in Nt and Nb.

Author Response

(The authors gave the same response as above.)

Reviewer 3 Report

Dear authors,

congratulations for the interesting article. What I miss in your article is SEM images of zinc coating after the drawing process both for wires drawn in conventional dies and hydrodynamic dies and a discussion about them  (in 3. Results and dicsussion). I believe it will improve the overall authority of the article.

In 2. Materials and Methods authors give only SEM equipment (S-3400 N-type Hitachi etc.), they should add here the other methods and equipment used, which now are in 3. Results and discussion (Form Talysurf 50e profilometer, standard PN-EN 10244-1 etc.). Also in 3.Results and discussion they should add SEM images of structure of the zinc coating AFTER the drawing process - both for conventional and multi-stage drawing process, and a discussion about structure differencies (zinc coating)

Best regards.

Author Response

(The authors gave the same response as above.)

Round 2

Reviewer 1 Report

You took most of my comments on board. Thanks. 

Therefore, as far as I am concerned, you paper is ready for publication. 

You could, however, have taken my suggestions more literally, for example, when I called for less "straightforward" figures, the idea was not to put exactly the same graphs as before under (above) the same caption!

Also, one micrograph look like not very much, on  such a topic!

Anyway, you might want to take on board the language points I highlighted in the attached document. 

Author Response

Thank You all for complex and accurate revisions of our manuscript.
